# Evaluation of near-infrared spectroscopy as a contactless method for health monitoring of resin-based coating materials applied to concrete surfaces

**Anri Watanabe** [1,2]*, **Masayuki Omiya**[3], **Makoto Sato**[3], **Hiromitsu Furukawa**[1], **Nobuko Fukuda**[1], **Hiroshi Minagawa**[4]

**1** Sensing System Research Center, National Institute of Advanced Industrial Science and Technology, Ibaraki, Japan, **2** AIST-TohokuU Mathematics for Advanced Materials Open Innovation Laboratory, National Institute of Advanced Industrial Science and Technology, Miyagi, Japan, **3** Fukamatsugumi Co. Ltd, Miyagi, Japan, **4** Graduate School of Engineering, Department of Civil and Environmental Engineering, Tohoku University, Miyagi, Japan

* a.watanabe@aist.go.jp

**Data Availability Statement:** All relevant data are within the manuscript.

## Abstract

The surfaces of concrete structures are often coated with protective materials to minimize corrosion and weathering-based deterioration. Therefore, it is important to monitor the aging of the coating materials and their overall condition to extend the service lifetime of the structure effectively. Near-infrared spectroscopy (NIRS) is a contactless, nondestructive, rapid, and convenient method for material characterization; therefore, it is useful for onsite inspection of coating materials. Hence, in this study, we attempt to determine whether NIRS can be used for simple inspection for health monitoring of organic resin-based coating materials. In addition to identifying different severities of peeling damage, we characterize the ultraviolet-induced deterioration of coating materials with different thicknesses using diffuse reflection spectra acquired in the near-infrared wavelength region. For independent comparison with the NIR spectra, the state of the coating materials on the mortar specimens was analyzed using a combination of Fourier-transform infrared spectroscopy and scanning electron microscopy, while the state of the underlying mortar specimens was analyzed using permeability and salt-water immersion tests. The results confirm that the NIRS could detect the degradation of coating materials at early stages of deterioration before their permeability had been affected. NIRS offers the possibility of intermittent monitoring of coating deterioration. In addition, because the NIR spectrometer is portable, it can help in inspecting high-rise areas and areas that are difficult to reach. Therefore, we believe that NIRS is a simple, safe, and inexpensive method for inspection of surface coating materials.

## 1. Introduction

For sustainable engineering, it is important to extend the life of infrastructure based on concrete structures. To this end, resin-based coating materials, such as acrylic, silicon, and epoxy,

**Funding:** The authors received no specific funding for this work.

**Competing interests:** The authors have declared that no competing interests exist.

are often applied to reinforced concrete structures to improve their durability [1], as these materials suppress the effects of aggressive external agents. For instance, as reported previously, applying resin to concrete surfaces and reinforcing bars suppresses the penetration of chloride ions, thus reducing corrosion of reinforcing bars [2, 3]. Conversely, corrosion begins when the resin protective film deteriorates [4]. In addition to highlighting the effectiveness of resin-based coatings in protecting against chloride attack [5], many studies have demonstrated that these resin-based coating materials suppress carbonation [6, 7], sulfuric acid attack [8], and frost attack [9]. These resin-based coating materials have been used in several concrete structures because they are cost-effective and they can extend the service life of a reinforced concrete structure, thereby eliminating the need of rebuilding the structure or large-scale repair.

Despite this protection, resin-based coating materials degrade in harsh environments, while their continued use in general outdoor environments gradually deteriorates their condition. The durability of different coating materials has been evaluated, and the results indicate that organic materials, such as acrylic, epoxy, and polyurethane, are more prone to deterioration than inorganic geopolymers [10, 11]. To ensure the effectiveness of resin-based coating materials in extending the service life of concrete structures, it is important to monitor their deterioration. The degree of deterioration of a structure with a long service period can be evaluated by intermittently checking the state of a resin-based coating material [12, 13]. The deterioration and lifetime of resin-based coating materials depend on the film thickness, number of years after application, coating layering, coating material, surface roughness, and sunshine direction. The typical methods for assessing the deterioration are visual inspection, hammering tests, and infrared thermography. However, recently, more advanced analytical techniques have been employed in a range of investigations. For instance, gas mass spectrometry and Fourier-transform infrared (FTIR) spectroscopy have been used for general qualitative and quantitative analyses of resin-based coating materials, while electrochemical methods have been used for time-dependent tracking of the deterioration of coating materials on metal surfaces [14]. Similarly, laser-induced breakdown spectroscopy (LIBS) has been proposed for the identification of fire-retardant/resistant coating in post-fire analysis [15]. At present, simple and immediate measurement methods are required in the construction field. Near-infrared spectroscopy (NIRS), a technique employed extensively in the fields of pharmacy [16–18], synthetic chemistry [19], petrochemistry [20], and food processing [21, 22] is a powerful tool for chemical analysis during and after production, and even useful for real-time monitoring. Also, visible and NIR spectroscopy is often used for prediction of physical and mechanical properties, supported by chemometrics procedures[23–25]. Chemical analysis using NIRS is similar to that using FTIR. In the mid-infrared region used in FTIR analysis, each functional group absorbs infrared light at several characteristic wavelengths (functional group region), appearing as sharp peaks, therefore it can be identified based on absorption wavelengths. The interpretation of NIR and IR spectra of polymer paint has been investigated [26]. The concentration of each functional group is determined by its peak intensity. Hence, infrared spectroscopy is capable of material identification, as there is a specific absorption spectrum for each substance. Quantitative evaluation of a material is possible in the NIR spectrum because the absorption peaks correspond to high-order harmonic and combination tones of the peaks in the mid-infrared region used in FTIR. However, there is a selectivity problem associated NIRS, because in the NIR region, absorption peaks derived from functional groups containing hydroxide overlap with each other. Nevertheless, NIRS is still the most suitable technique for nondestructive analysis of thick samples, and, in locations with extreme operating environments, it can be conducted remotely [27] because analysis is performed rapidly in place, with no sample preparation. Unlike inspections that require samples to be extracted from objects, NIRS offers the potential for rapid, nondestructive analysis of an entire coating surface, and thus, this technique will be beneficial for construction and maintenance.

In this study, we propose NIRS to investigate insensible chemical deterioration of the painted surface of concrete. Early detection of the deterioration makes the strategic maintenance management more straightorward, which is thought to be important for longer service life. NIRS is able to identify any signs of deterioration even before the functionality of the painted surface is lost. As measurement objects, water-based coating agents, which have been widely used because they are harmless to the human body, and mortar specimens which do not contain aggregates to simplify the system, were chosen. The effect of slight chemical deterioration on the durability performance of the coated mortar specimens is assumed to vary depending on the coating thickness. Therefore, we prepared mortar specimens with three different thicknesses of resin-based coating and performed weathering tests using UV irradiation to evaluate chemical deterioration. After the deterioration, NIRS measurements were carried out in a contactless manner to investigate the differences between the NIR spectra of samples with and without the weathering test. To evaluate the validity of the deterioration detection by NIRS, the deterioration state of the resin-based coating material itself was confirmed with conventional FTIR and the surface fine structure was examined by scanning electron microscopy (SEM). In addition, even when the deterioration is detected by NIRS, it is necessary that the degree of deterioration of the resin-based coating material does not cause the loss of functionality (because our aim is to identify the early signs of deterioration with NIRS). To investigate the effect on the performance of the mortar specimens beneath the coatings, water permeability test and salt-water immersion test were performed.

Furthermore, we obtained the NIRS spectra of concrete plates coated with different combinations of the layers of the resin-based coating material for assessing whether different degrees of peeling could be identified. We believe that this will be revolutionary to inspect the entire coating surface in a short period of time in a nondestructive manner, avoiding the need to inspect the chemical deterioration of the resin-based coating material after long period of time.

## 2. Materials and methods

### 2.1. Preparation of specimens

Four mortar specimens, consisting of a mixture of ordinary Portland cement and fine aggregates (crushed sand from Kakegawa, Shizuoka prefecture, Japan), were prepared under the same conditions, i.e. the water-to-cement ratio was set at 0.6:1, while the sand-to-cement ratio was set at 3.0:1.0. After mixing and casting the mortar into a mold with internal dimensions of approximately 300 mm × 200 mm × 40 mm to produce a mortar plate, the mold with the mortar was sealed and cured at room temperature for 28 days. Next, the mortar plate was demolded and subsequently coated by applying acrylic resin, which is the main agent of the coating material, on the surface of the mortar plate in contact with the bottom of the mold, and was used for weather resistance evaluation. The coating was cured in air for more than 24 h. At actual construction sites, in addition to the main agent, base materials and top coats are applied to the concrete surface for protection. However, because the aim of this study is to assess the feasibility of nondestructive testing of the condition of a surface coating material, only the main agent was applied to the mortar to minimize complicated interactions in the system. The main agent (KIKUSUI Chemical Industries Co., Ltd., 10ELR-4) was composed of an acrylic-styrene resin emulsion (9.7 wt%), $CaCO_3$ extender pigment (77.1 wt%), water (10.6 wt %), and additives (2.6 wt%), which consist of an anti-foaming agent, a dispersant, and a thickener. We considered four different types of samples in our investigation: one without a resin coating (hereafter referred to as the non-coating sample), one with a single coating of resin applied with wool rollers (thin coating sample), one with a single coating of resin applied with mastic (normal coating sample), and one with coatings of resin applied twice with mastic

(thick coating sample). The thicknesses of the normal and thick coating samples were measured to be 1.5 mm and 1.9 mm, respectively, using a ULT-5000 ultrasonic thickness gauge. However, the thickness of the thin coating sample was too small (< 0.5 mm) for precise measurement using this tool. Several cylindrical mortar specimens (diameter: 25 mm, height: ~40 mm) were collected from these mortar plates using a coring machine and were subjected to UV irradiation. The NIR spectra of these cylindrical samples before and after UV irradiation were obtained as described in Section 2.2, and the results of this analysis were validated using conventional FTIR spectroscopy. Finally, the deterioration of the internal mortar was examined using a combination of permeability and salt-water immersion tests (performed with separate cores). Totally, 2 (before/after UV irradiation) × 4 (coating thickness) × 5 (kinds of measurements, in case of NIRS, FTIR, Permeability test, Salt-water immersion test, SEM) = 40 samples were prepared.

Samples for the peeling characterization tests were prepared as follows. First, polytetrafluoroethylene (PTFE) diffuse reflector sheets (Thorlabs, Inc., PMR10P1) of dimensions 33 cm × 33 cm × 0.75 mm were cut into quarters. The epoxy base material, the main agent (used in the weather resistance test), and the acrylic topcoat of the resin-based coating were subsequently applied to each sheet, such that the specific spectra of each individual layer could be identified. The base material (KIKUSUI Chemical Industries Co., Ltd., 06SPE-10) was completely made of modified acrylic resin emulsion. The topcoat (KIKUSUI Chemical Industries Co., Ltd., 11BTU-9) was composed of an acrylic urethane resin emulsion (62.1 wt%), $TiO_2$ as white pigment (16.9 wt%), water (9.0 wt%), additives (11.5 wt%), seaweed proof/mildew proof agents (0.5 wt%), and an aqueous color dispersion (quantum placet). The series of coatings, base material, main agent, and topcoat used in this study meet the requirement of JIS (Japanese Industrial Standards) A6909 [28]. Then, we prepared four types of samples using commercial concrete slabs (NXstyle 3-30SD, 300 mm × 30 mm × 300 mm). The first sample (undamaged sample) consisted of a concrete slab coated in order with the base material, the main agent, and the topcoat; the second sample (moderate damage sample) consisted of a concrete slab coated with the base material and the main agent, to model the case in which only the topcoat has peeled off; the third sample (severe damage sample) consisted of a concrete slab coated with only the base material, modeling the case in which the top coat and the main agent have peeled off; and the final sample consisted of a concrete slab coated with the base material and the top coat, modeling the case in which the main agent is erroneously omitted from the coating. The thicknesses of the base material, main agent, and top coat were 0.1 mm, 1 mm, and 0.25 mm, respectively, for each sample. The total number of the samples for the peeling characterization was four.

## 2.2. Acquisition of near-infrared absorption spectrum

NIR spectra for the four types of mortar specimens (before and after UV irradiation), PTFE diffuse reflector sheets, and four types of concrete samples described above were obtained using a near-infrared Fourier-transform spectrometer (FT-NIR Rocket, ARC Optix, Neuchatel, Switzerland, λ = 900–2600 nm, spectral resolution of 4 cm$^{-1}$). In this experiment, a halogen lamp (HSH-30, Fintech. Co. Ltd., Kobe, Japan) was used for illumination of the measured objects, and its voltage and current were set at 3.00 V and 3.98 A, respectively. The light source was preheated for 30 min to ensure stable illumination. The experimental setup for obtaining the NIR spectra is illustrated in Fig 3 of [29]. Briefly, the light source was oriented at an angle of 30° to the samples, and the probe connected to the spectrometer via an optical fiber was placed above the sample, such that diffuse reflection can be introduced to the spectrometer, effectively avoiding normal reflections. Each sample was placed 200 mm from the light source, and 135 mm from the probe. The diffuse-reflected light was measured by installing a mask

made of a velvet anti-reflection sheet with a hole 16 mm in diameter. This mask was then placed on the surface of the sample to equalize the area to be measured. A reference spectrum was obtained using a diffusion plate (reflectance of 99%). Finally, the gain of the amplifier was set at extreme, while the number of averaging (accumulating) spectra was set at 10 using the supplied software.

## 2.3. Ultraviolet irradiation

The mortar specimens, prepared as described in Section 2.1, were subjected to UV irradiation to accelerate the deterioration of the surface coating material. To ensure that only the surface coating material was irradiated, the side and the bottom of the mortar specimens were shielded with aluminum foil. UV irradiation was performed for 392 h using a metal halide weather resistance evaluation device (i-super UV tester SUV-W161, Iwasaki Electric) with the irradiation intensity set at 1500 W/m$^2$ (± 10%) at 300–400 nm in wavelength. According to JIS D0205, average annual UV radiant exposure in Japan is 4500MJ/m$^2$ [30], and its composition is 300–400 nm (6.8%) [31], 400–700 nm (44.6%), 700–3000 nm (48.6%). Therefore, the annual radiant exposure in the wavelength range of 300–400 nm is 4500 x 6.8% = 306MJ/m$^2$. The irradiation of 1500W/m$^2$ for 392 h = 392 × 60 × 60 s = 1 411 200 s in this experiment corresponds to 1500 × 1 411 200/ (306 × 10$^6$) = 6.9 years outdoor UV exposure. During these tests, the temperature was maintained at 63±2˚C. To model rainfall in the outdoor environment, pure water was sprayed on the surface of the mortar specimen for 1 min every 60 min. Although the humidity in the chamber increased to ~100% immediately after spraying pure water, the humidity in the chamber was maintained at 50±5% at all other times of operation.

## 2.4. FTIR spectroscopy

Because the specific absorptions of multiple functional groups may overlap in the near-infrared absorption spectrum, the spectral assignment is difficult without confirmation by conventional methods. Thus we performed FTIR spectroscopy in the functional group region to enable their identification. The spectra of NIR and FTIR were compared, under the assumption that the major difference of spectra in the NIR before and after deteriorating reaction would correspond to that in the IR region. Here, diffuse-reflectance IR spectra were obtained for the normal coating samples before and after UV irradiation, and were shown expressed as a Kubelka-Munk function (K/S value). The FTIR spectrometer (Thermo Fisher Scientific K.K, Nicolet iS50) had a diffuse reflection unit and a deuterium triglycine sulfate detector; the resolution was set at 2 cm$^{-1}$ and the spectrum for each sample was collected 256 times.

## 2.5. Permeability test (JIS standard)

It is well known that the protective organic paintings are deteriorated by mainly UV irradiation during weathering test. This photo-oxidative deterioration decreases water protection ability; therefore, the permeability of organic coatings generally increases with longer period of exposure to UV irradiation [32, 33].

 A permeability test in accordance with "JISA6909 (2014) finishing coating material for construction: permeability test B method" [28] was performed to determine whether the deterioration of the coating material (main agent) affected the mortar under the coating. We cut two mortar specimens with normal coating (one each from before and after UV irradiation) in the direction perpendicular to the cylinder's axis. An uncoated portion from the middle of the mortar specimen before UV irradiation was used as the reference for determining the rate of water absorption by the unprotected mortar prior to the effects of weathering. We avoided using the lower part of the sample opposite the coating surface as a reference because it

contained the casting surface. Thus, the density of the mortar in this region may have been too small to enable accurate measurement.

For pre-treatment, the mortar specimens that had been cut were dried at 40°C for 7 days until the change in mass per day had converged to ~0.1%, indicating that drying was complete. Following this, the permeability test was conducted according to the description in JISA6909. Since adhesion to the glass siphon specified in JISA 6909 was difficult owing to the shape of the mortar specimens, we replaced this siphon with an acrylic cell made by cutting an acrylic pipe to a suitable length. The mortar specimens were loaded into the cell by gluing their sides to the inner surface of the cell. Once loaded, a measuring pipette (0.05 mL) with a 5-mL yield was connected to the acrylic cell using a silicon stopper. The cell and the measuring pipette were then filled with water to a height of 250 mm from the top surface of the sample, and the permeability test was started after the liquid level had been marked. The volume of water permeated through the mortar specimens was measured by the change in the water level 24 h after the start of the test, using the pipette as the scale. The test was carried out at a temperature of 21–22°C.

## 2.6. Salt-water immersion test

We decided to do salt-water immersion test in order to confirm the protective function of the coating was sustained even after UV irradiation. The procedures of this test including pre-treatment were according to JSCE (Japan Society of Civil Engineers) standards [34]. The analysis of the spatial distribution of chloride ion was then detected by electron probe microanalyzer (EPMA) [35].

Before salt-water immersion, a water saturation pre-treatment was conducted on the four mortar specimens (non-coating, thin coating, normal coating, and thick coating samples) after UV irradiation and the normal coating sample before UV irradiation. Here, the specimens were submerged in pure water in a container, followed by vacuum-degassing for 1 h. After this degassing period, atmosphere was released into the container, and the specimens were maintained in submersion for two days. After saturation pre-treatment, the specimens were removed from water, and dried for ~4 h. The uncoated surfaces of the specimens were subsequently sealed with epoxy resin, which was cured by incubating the specimens in a wet environment for 3 days. The salt-water immersion tests were performed after the epoxy resin had been cured.

To eliminate the influence of drying caused by the epoxy resin coating process, the specimens were first immersed in pure water for 24 h. After this, the water adhered to the surfaces of the specimens was removed, and they were subsequently immersed for 14 days in a 0.5 mol/L NaCl solution maintained at ~20°C. Once removed from the NaCl solution, the specimens were cut in half across the cross-sectional direction, and were dried under the indoor environmental condition. Finally, sample preparation was completed by mirror polishing the cutting surface. To investigate the depth of penetration by chloride ions (Cl⁻) in the prepared samples, we performed elemental mapping at an acceleration voltage of 15 kV using an electron probe microanalyzer (EPMA) (JXA-8200, NEC). Cl⁻ ions were measured at a sample current of 200 nA, using a probe diameter of 50 μm, a pixel size of 100 μm, and an acquisition time of 40 ms/pixel. We used a halite crystal (Cl = 60.7%) as the standard sample in our analysis, and PET as the spectroscopic crystal. Finally, we estimated the Cl⁻ concentration in a sample using a proportional conversion method [35].

## 2.7. Electron microscopy

In order to know the detail surface structure, we conducted SEM measurement for the normal coating specimens before and after UV irradiation. The surface of these specimens were not coated by conductive film, such as gold or carbon, to prevent charge-up. Hitachi SU5000 with

an acceleration voltage of 15 kV was used, under the vacuum level of 50 Pa. The accumulation was 30 times. The elemental composition of the mortar specimen was analyzed using energy dispersive X-ray (EDX) spectroscopy, conducted with the microscope's spectrometer accessory (Oxford Instruments X-Max$^N$ 50 with Aztec analysis software).

# 3. Results and discussion

## 3.1. NIRS measurements for mortar specimens with main agent of coating material

Fig 1 shows the absorbance spectra of the non-coating, thin coating, normal coating, and thick coating samples. Here, each spectrum has been shifted vertically to make the peaks easier to distinguish. The characteristic peaks of the main resin agent appear in the wavenumber ranges of 4180–4500 cm$^{-1}$, 4535–4725 cm$^{-1}$, and 5550–6055 cm$^{-1}$. The large peaks at 4264 cm$^{-1}$ and 4338 cm$^{-1}$ correspond to a C–H bond [36]; the absorption of this bond increases with thicker coating. This behavior indicates that the thickness of the resin coating can be determined using NIRS. We also got the information from thin coating; the spectrum for this thin coating specimen shared some characteristics with those for the normal coating and non-coating samples. Specifically, in addition to the absorption peaks characteristic of the coating, a sharp peak was observed at 7143 cm$^{-1}$, which was attributed to calcium hydroxide present on the mortar surface [37]. This peak proves the uneven coverage of the resin agent on the quite thin coating sample; explaining why peaks for both the mortar and the resin could be observed. The spectral mixing may also be due to the fact that light has passed through the coating layer of the main agent. Thus, we suggest that this behavior can be used to determine if there are any uncovered spots on concrete surface, or if the thickness of the resin agent needs to be increased.

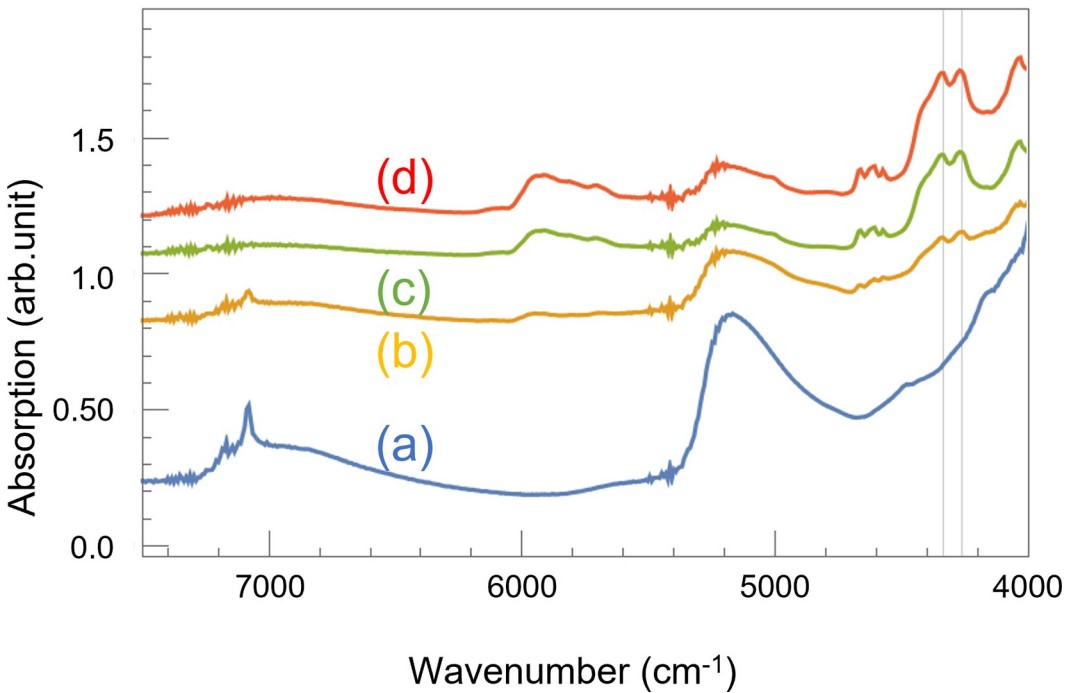

**Fig 1. Absorption spectra of the resin-coated mortar specimens in the near-infrared region before UV irradiation.** Blue line (a): non-coating; yellow line (b): thin coating; green line (c): normal coating; red line (d): thick coating. Grid lines indicate 4338 cm$^{-1}$ and 4264 cm$^{-1}$.

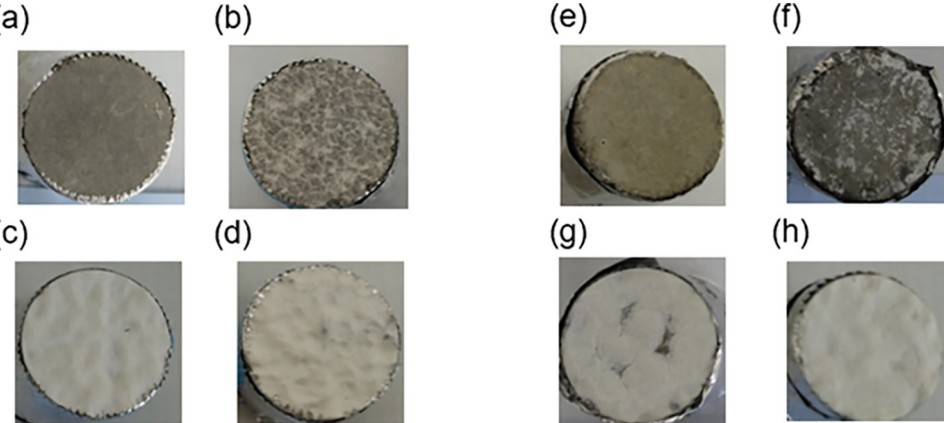

**Fig 2.** Photographs of the surfaces of the (a) non-coating, (b) thin coating, (c) normal coating, and (d) thick coating specimens before UV irradiation. Photographs of the surfaces of the (e) non-coating, (f) thin coating, (g) normal coating, and (h) thick coating specimens after UV irradiation. The sides of each core were protected with aluminum foil, to restrict the UV radiation to the coated surface of the mortar specimen.

Fig 2 shows photographs of mortar specimens before (Fig 2A–2D) and after (Fig 2E–2H) UV irradiation conducted to simulate age-based deterioration. The amount of UV irradiation absorbed by the specimens is equivalent to seven years' of sunlight, which is less than the nominal service life (10 years) of the coating material used in this study. Hence, although some coating material was peeled off the mortar surfaces after irradiation, most of this coating material (white paint) with normal and thick coating was retained.

To investigate whether NIRS could detect the aging deterioration of a coating material, we obtained the absorbance spectra of the UV irradiated specimens in the NIR region, as shown in Fig 3. Here, each graph has been shifted vertically to make the peaks easier to distinguish. A comparison between Figs 1 and 3 indicates that UV irradiation suppressed all the characteristic peaks of the coating material (4180–4500 cm$^{-1}$, 4535–4725 cm$^{-1}$, 5550–6055 cm$^{-1}$). The reduction in the absorption intensity of the peak at 4338cm$^{-1}$ is particularly stark compared to the peak at 4264 cm$^{-1}$, indicating that the C-H bond of $CH_2$ antisymmetric stretching is more easily broken than that of $CH_2$ symmetric stretching [36] by UV irradiation. At first glance, a reasonable assumption would be that the differences between Figs 1 and 3 are because the UV irradiation peeled off part of the coating material. However, the visual inspection summarized in Fig 2 indicates that this simple explanation seems to be valid only for the thin coating specimen. The UV irradiation removed most of the paint from the thin coating sample, as shown in Fig 2F. Consequently, the shape of the spectrum for this sample resembled that of the non-coating sample, as the absorption peaks for the paint were suppressed. Conversely, although visual inspection indicates that most of the coating material remained on the specimen with thick coating after UV irradiation (Fig 2H), there are clear differences between the spectrum for this specimen in Fig 3 and the one in Fig 1 (red line). Finally, the only noticeable difference between the spectra for the noncoated specimen in Figs 1 and 3 (blue solid line) is the disappearance of the absorption band at 7143 cm$^{-1}$. We consider this disappearance to be caused by the carbonation of the mortar surface resulting from intermittent wetting and drying of the samples to simulate rainfall. These results indicate that NIRS can detect coating deterioration more precisely than visual inspection.

It is noteworthy that the prominent braod peaks of water appeared in Fig 3 at around 5200 cm$^{-1}$ and 7000 cm$^{-1}$, which may be because the large amount of water was applied to the samples by the shower during the weathering test.

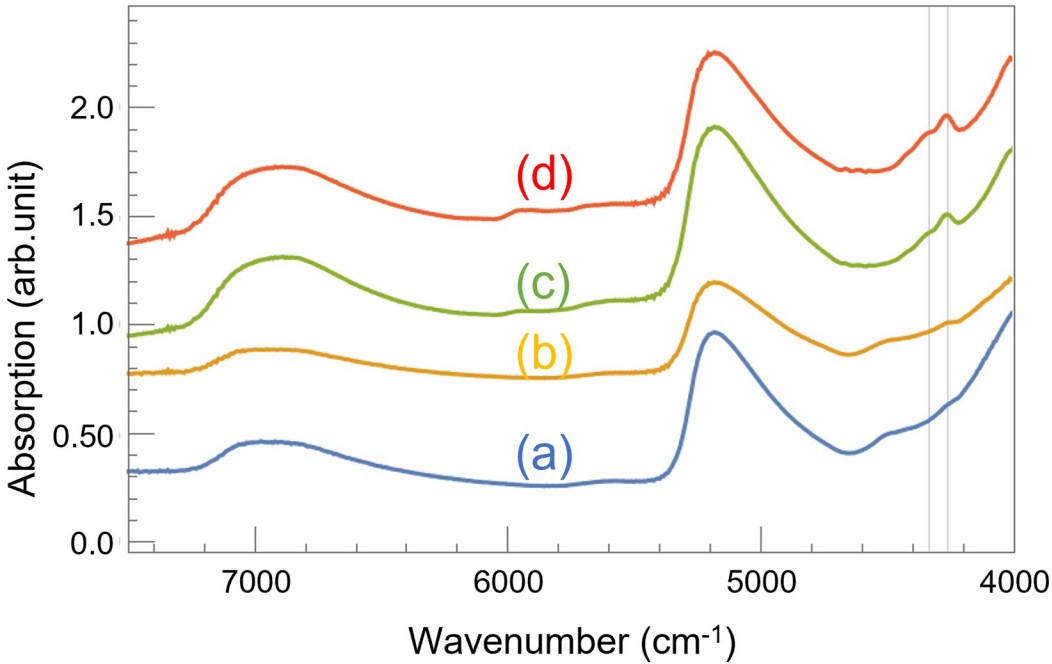

**Fig 3. Absorption spectra of the resin-coated mortar specimens in the near-infrared region after UV irradiation.** Blue line (a): non-coating; yellow line (b): thin coating; green line (c): normal coating; and red line (d): thick coating. Grid lines indicate 4338 cm$^{-1}$ and 4264 cm$^{-1}$.

## 3.2. FTIR measurements

To verify the accuracy of the comparisons made using NIRS, we obtained IR spectra of the specimen with normal coating before and after UV irradiation, as shown in Fig 4. This procedure makes it possible to confirm the specific peaks present in the NIRS spectra, as conventional FTIR spectroscopy in the functional group region avoids the overlap between the absorption bands in the NIR region. Here, the plots have been shifted along the vertical axis for easy observation of the change in the intensities of the peaks. As measurement was conducted using diffuse light, the intensity of each peak was standardized to that of the peak at 2512 cm$^{-1}$, where no significant absorption should be usually observed. In these plots, we observed peaks in the 1200–1300 cm$^{-1}$, 1730 cm$^{-1}$, and 2800–3000 cm$^{-1}$ wavenumber regions, corresponding to absorption by C-O bonds in esters, carbonyl groups (C = O), and alkyl groups (C-H stretching), respectively [26, 38]. The magnitudes of all these peaks decreased after UV irradiation. Here, the decrease in the magnitude of the peak at 2950 cm$^{-1}$ is greater than the decrease in the magnitude of the peak at 2870 cm$^{-1}$, which is believed to correspond to the behavior of the absorption band at 4338 cm$^{-1}$ in the NIRS. The peak intensity ratios for 2950 cm$^{-1}$ to 2870 cm$^{-1}$ in FTIR (normal coating) were 1.15 (before UV irradiation) and 0.638 (after UV irradiation), where the peak intensity was measured from the intensity at 3130 cm$^{-1}$, the skirt of the peak. While, the peak intensity ratio for 4338 cm$^{-1}$ to 4264cm$^{-1}$ in NIR (normal coating) were 0.966 (before UV irradiation) and 0.608 (after UV irradiation), where the peak intensity is measured from the intensity at 4480 cm$^{-1}$, the skirt of the peak. Although the changes of peak intensity cannot be compared directly since the sensitivity of each wavenumber is different between NIR and IR spectrometers, the peak intensity ratios for 4338 cm$^{-1}$ to 4264cm$^{-1}$ in NIR and that for 2950 cm$^{-1}$ to 2870 cm$^{-1}$ in IR were reduced to about 63% and 55% respectively, due to UV irradiation. Based on the location of these peaks, this decrease

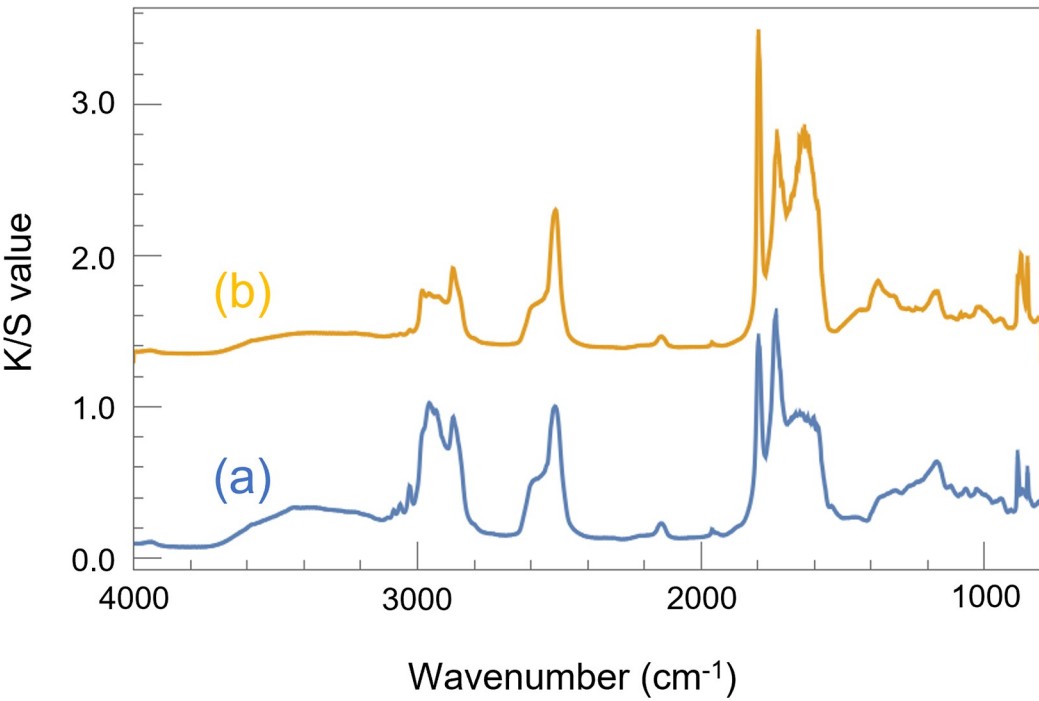

**Fig 4.** FTIR spectrum of the surface of the specimen with normal coating before UV irradiation ((a)blue line) and after UV irradiation ((b)yellow line).

appears to indicate a C-H bond cleavage of $CH_2$ in the linear alkyl group. Moreover, it is assumed that the inorganic particles of $CaCO_3$ caused the larger peak at 1500–1700 cm$^{-1}$ in the spectrum after UV irradiation owing to the alkyl group cleavage of acrylic-styrene resin emulsion [39].

### 3.3 Permeability test

For additional verification of the accuracy of the NIRS evaluation, we investigated the properties of the mortar specimens underneath the resin coating. The results of the permeability test of the specimen with normal coating are shown in Fig 5. Here, approximately l mL of water permeated through the intermediate section of the mortar specimens, regardless of whether they had been stimulated with UV irradiation. UV irradiation had a similarly negligible effect on the permeability of the surface samples. Here, the unirradiated specimen permeated 0.25 mL of water, while the irradiated specimen permeated 0.20 mL of water. This reduction in the volume of water lost by the surface mortar specimens was consistent with their smaller mass. In addition, the irradiated surface sample seemed slightly browned. This change in appearance might suggest carbonation. To verify this, the sample was cut in the axial direction, and its cross section was sprayed with phenolphthalein solution. The resulting test produced no evidence of carbonation, confirming that, for a normally coated sample, the UV irradiation conditions used in this study do not affect the mortar specimen.

### 3.4. Salt-water immersion test

Similar to Section 3.3, the conventional inspection of the salt-water immersion test is performed as a reference test of NIR spectroscopy, which proves the protection from the external substance both before and after the UV light irradiation. The results of the salt-water

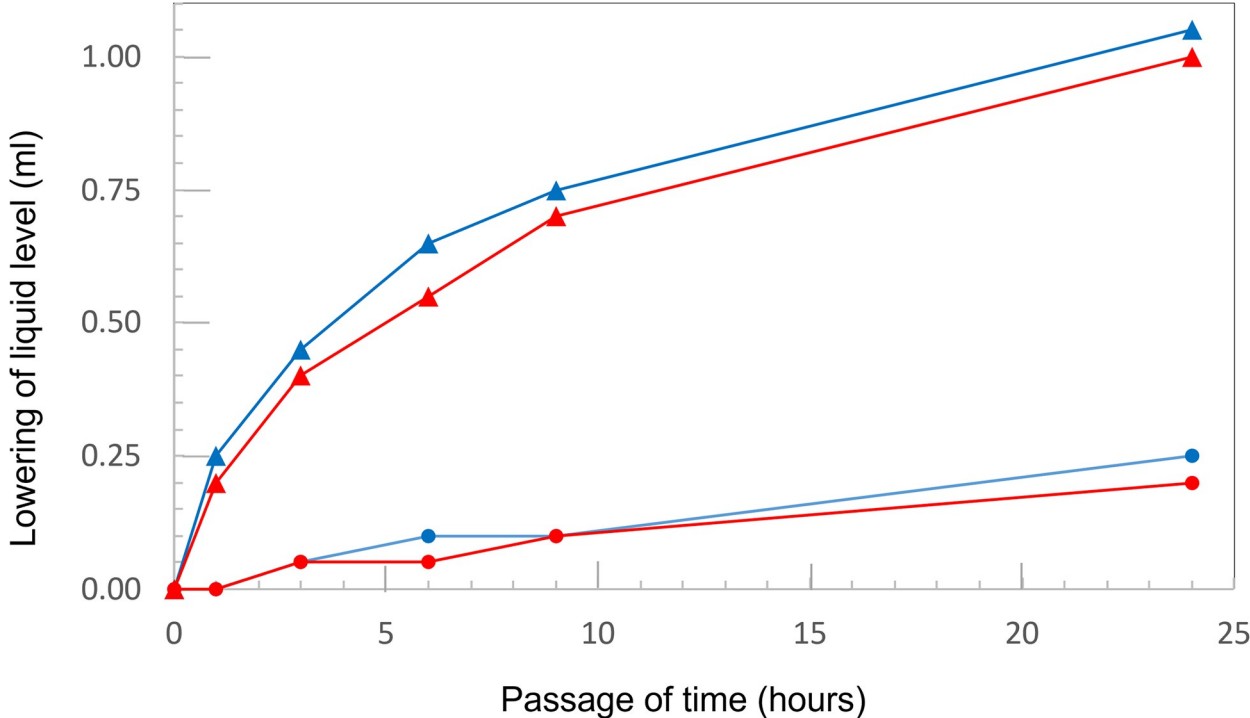

**Fig 5. Permeability of the specimens with normal coating before (blue line) and after UV irradiation (red line).** Here, the circle indicates measurements taken using the coated surface of the mortar specimen, while the triangle indicates measurement taken using the inner portion of the specimen without a coating.

immersion test conducted to examine the penetration of chloride ions into the mortar specimens are shown in Fig 6. For the normal coating sample, there were minimal differences in the penetration depth of the chloride ions before (Fig 6A) and after UV irradiation (Fig 6B-3), indicating that the aging deterioration of the coating was not significant enough to affect the internal state of the mortar. We also observed that the penetration depth of the chloride ions was shallower in the case of the specimen with thick coating (Fig 6B-4), confirming the protective effect of the surface coating material. However, the penetration depth of the chloride ions was shallower in case of the noncoated specimen (Fig 6B-1) than in case of the specimen with thin coating (Fig 6B-2). This result suggests that a protective $CaCO_3$ film was formed on the surface of the noncoated specimen as a result of mortar carbonation.

### 3.5. Investigation of fine structure of the surface of mortar specimens

The micro structural information was obtained as supplemental data to explain the NIR measurement. To verify our hypothesis that the $CaCO_3$ particles appered on the surface of the normal coating specimen after the UV irradiation, we conducted plain-view SEM measurements on the surface of specimens with normal coating before and after UV irradiation (Fig 7). It was confirmed by near-infrared spectroscopic nondestructive analysis (1 second or less per single measurement) that some chemical deterioration had occurred in the coating of the sample surface by UV irradiation. By FTIR measurement, it was identified that the bonds that deteriorated in the coating was a C-H bond of alkyl group. Further, EPMA and water permeability tests on the coated cement paste specimen revealed that there was no difference in the salt penetration depth and water permeability amount into the cement paste specimen, and this is considered to be owing to the fact that there is almost no deterioration in the physical

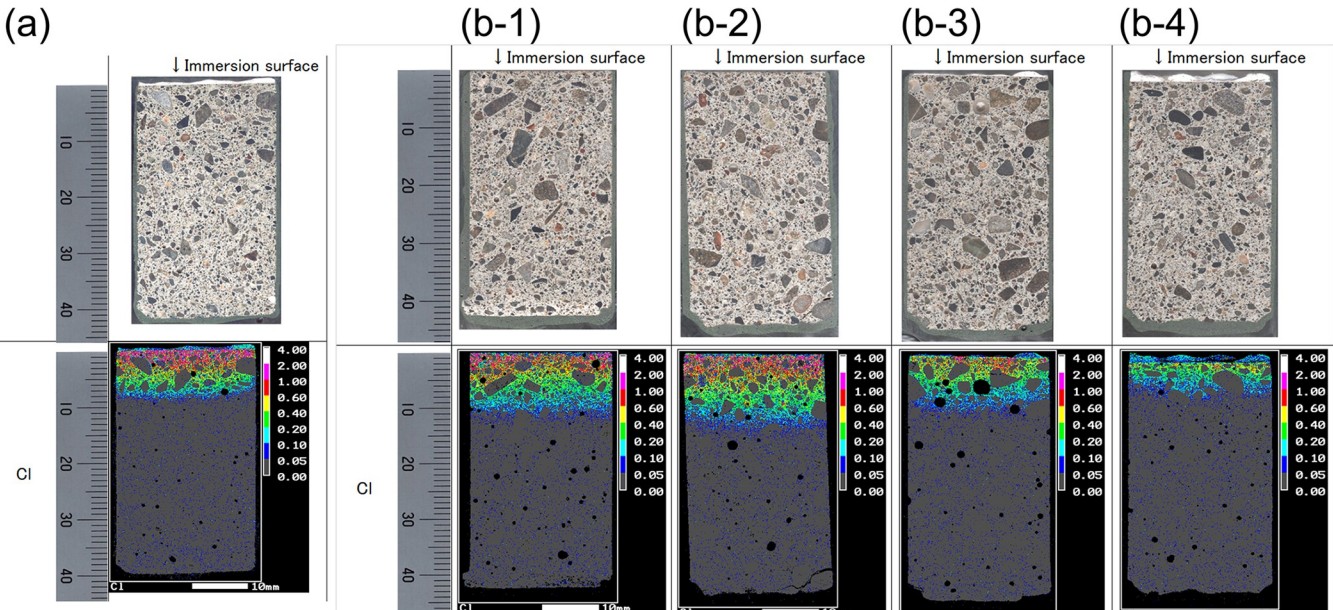

**Fig 6.** Electron probe microanalyzer image of chloride ion penetration in the resin-coated mortar specimens: (a) specimen with normal coating before UV irradiation, (b-1) noncoated specimen after UV irradiation, (b-2) specimen with thin coating after UV irradiation, (b-3) specimen with normal coating after UV irradiation, and (b-4) specimen with thick coating after UV irradiation.

performance of the coating due to UV irradiation equivalent to about 7 years. Here, to analyze the shape and components of the surface layer, a backscattered electron image, a secondary electron image, and an EDX mapping image were acquired using SEM for samples of normal coating, before and after UV irradiation. From the backscattered electron image, it can be seen that physical changes such as cracks have not occurred owing to UV irradiation (Fig 7A and 7F). According to the enlarged figure, particles of similar size exist on the surface layer (Fig 8A and 8C, before and after the UV irradiation, respectively). Focusing on the secondary electron images (Fig 8B and 8D), it is assumed that particles are coated with something before UV irradiation; however, the particles appear to be exposed after UV irradiation. Before UV irradiation, a large amount of C exists on the surface of the particle and between the particles, and Ca is distributed under it, according to the EDX mappings of C, Ca, and O (Fig 7C–7E). After UV irradiation, the surface appears to have little C, while much Ca and O is distributed (Fig 7H–7J). Especially, the composition of this coating agent was set to have a structure in which $CaCO_3$ exists as granular crystals, and the acrylic resin existed in the inter granular space and also on the surfaces to cover them. However, UV irradiation breaks the C-H bond of the acrylic resin, and it is assumed that the damaged acrylic resin was washed off by a shower corresponding to the rainfall conditions, which is why the $CaCO_3$ granular crystals were exposed after UV irradiation. Because the $CaCO_3$ crystals remain intact, no deterioration was observed in terms of salinity penetration depth or water permeability, but it is considered to be an early deterioration of the coating film. These SEM-EDX images critically showed the validity of the optical non-destructive measurement, i.e. near-infrared spectroscopy can be said to have detected acrylic resin failure as initial deterioration by UV.

From the above, the difference was detected in the spectral shapes in NIRS spectra between specimens with surface coating before and after UV irradiation, even when visual inspection and moisture and chloride penetration tests did not indicate significant deterioration. This indicates that NIRS can non-destructively identify the early stages of coating degradation and

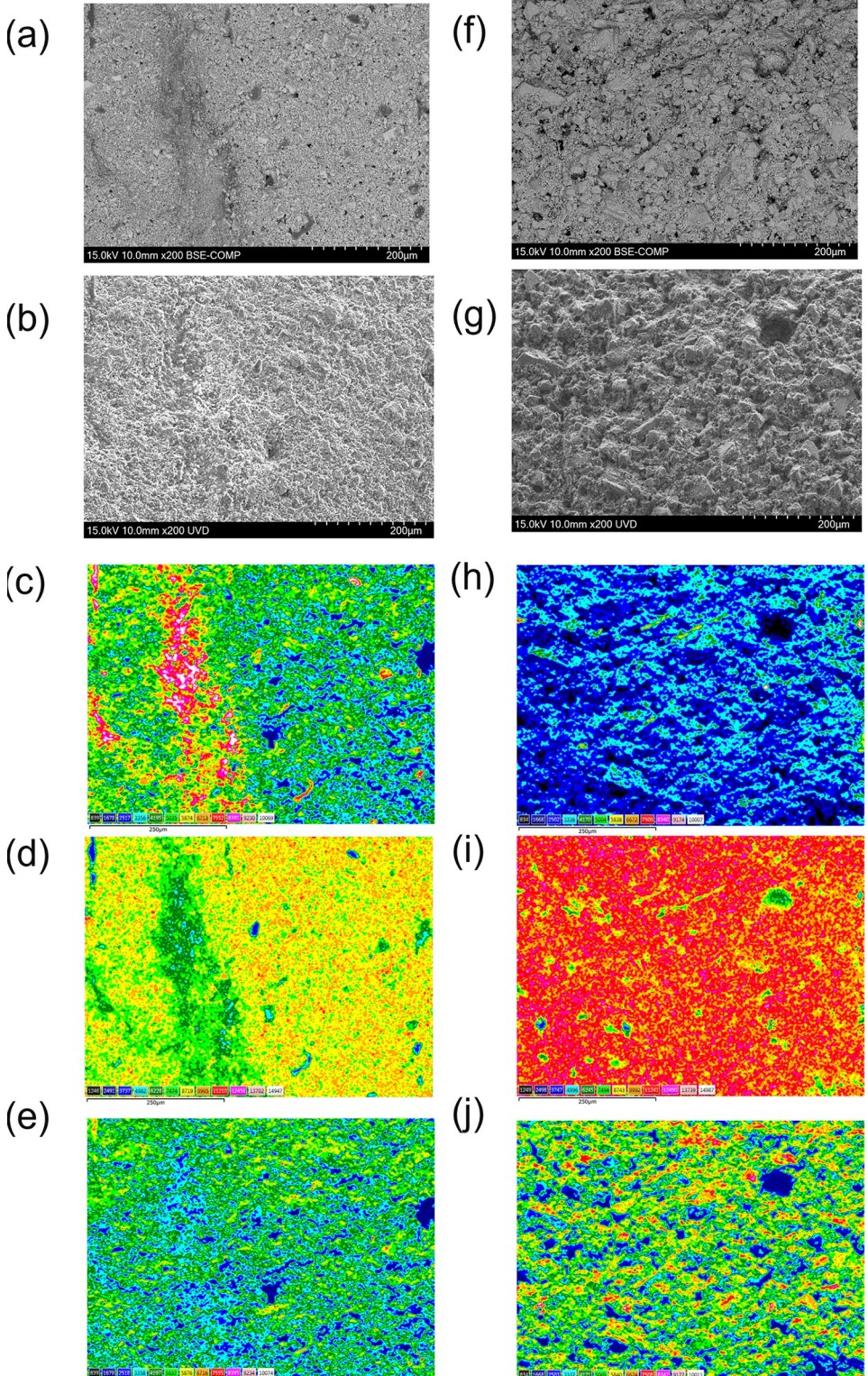

**Fig 7.** SEM and elemental EDX mapping images before (a)-(e) and after (f)-(j) UV irradiation. (a) and (f) are backscattered electron images. (b) and (g) are secondary electron images. EDX mapping images of C (c) and (h), Ca (d) and (i), and O (e) and (j). Color scale indicate the intensity of characteristic X-ray (arbitrary unit).

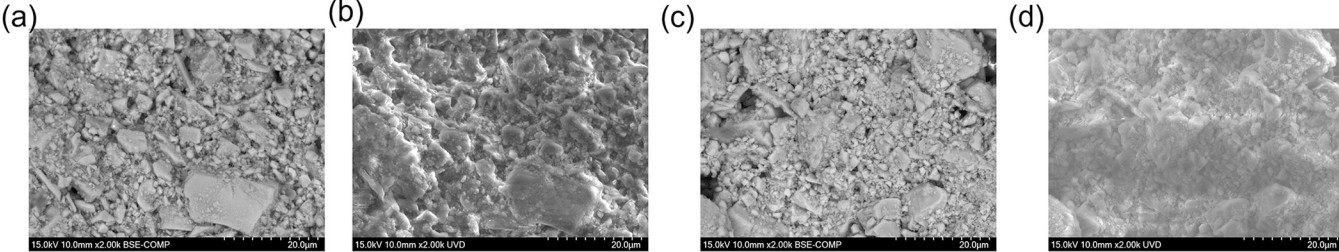

**Fig 8. The enlarged (high magnification) image of Fig 7.** (a) and (b) before UV irradiation, (c) and (d) after UV irradiation. (a), (c) backscattered electron image, and (b), (d) secondary electron image.

increase the possibility of preventive maintenance of concrete structures before coating degradation becomes serious. In the future, it is expected that the NIRS technique can be simply used by engineers in the field without sending samples to labs for SEM-EDX and other destructive measurements.

## 3.6. NIRS measurements for commercial concrete slabs with multi-layer-coatings

Typically, concrete surfaces are sequentially coated with base, main, and topcoat resins. To verify that NIRS could be used for the detection of missing layers, we measured the spectra of each material individually, as shown in Fig 9. Here, each graph was shifted vertically for easier identification of their specific features. There are clear differences in the shape of each spectrum, suggesting that, in principle, NIRS can be used to detect missing layers.

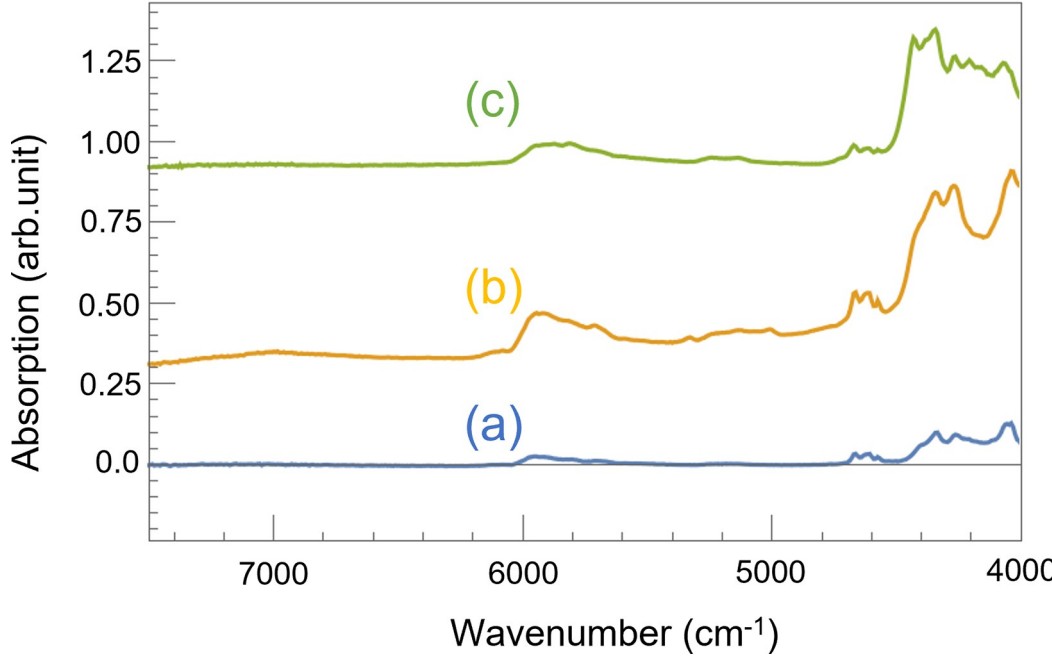

**Fig 9.** Absorption spectrum of the base material (blue) (a), main agent (yellow) (b), and topcoat (green) (c), of the acrylic coating in the near-infrared region.

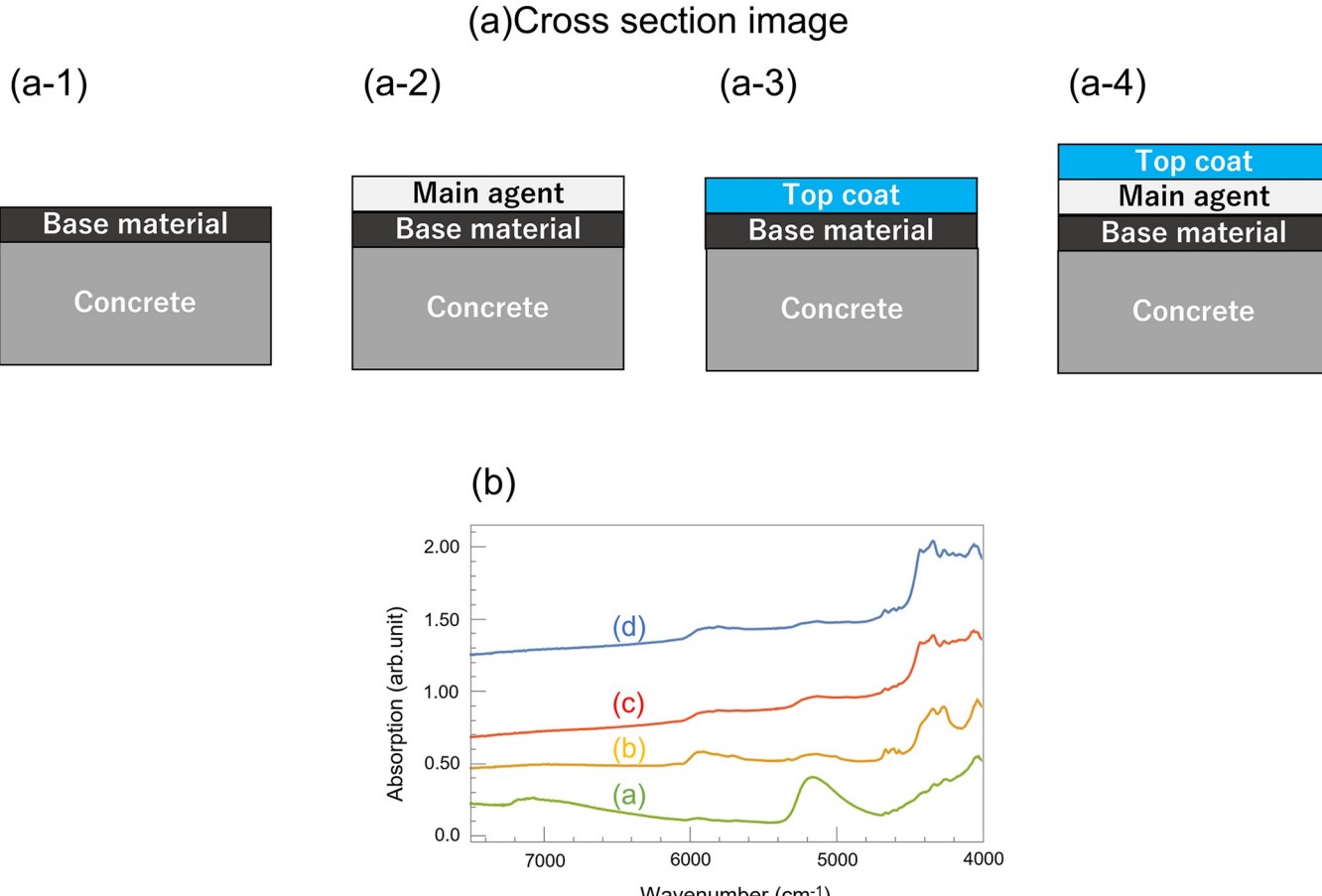

**Fig 10.** Schematic diagram of paint layers on a concrete plate modeling: (a-1) severe peeling damage (only the base material remains on the structure), (a-2) moderate peeling damage (the base material and the main agent remain), (a-3) erroneous application (no main agent is included in the resin), and (a-4) an undamaged coating (all layers applied correctly). (b) Absorption spectra of (a-1), (a-2), (a-3), and (a-4) in the near-infrared region (green (a), yellow (b), red (c), and blue (d), respectively).

Fig 10 shows the absorption spectra of the commercial concrete samples prepared as described in Section 2.1. Each graph was shifted vertically, for easier comparison of their characteristics. Here, the spectrum for each sample reflects that of the material currently at its surface. Thus, the spectrum for the moderately damaged sample (Fig 10A-2) is identical to that of the main agent, and the spectrum for the undamaged sample (Fig 10A-4) is identical to that of the top coat. The spectrum for the severely damaged sample (illustrated in Fig 10A-1) is conbination of those of the base material and non-coating specimen because the light has passed through the thin coating layer of base material. This indicates that NIRS is able to detect some forms of peeling damage. However, it was unable to identify the case when the main agent was erroneously omitted (Fig 10A-3).

## 4. Conclusion

In this study, we evaluated the feasibility of employing near-infrared spectroscopy for nondestructive analysis of coating deterioration. Here, weathering effects were simulated using UV irradiation. We observed differences in the NIRS spectra of unirradiated and irradiated mortar specimens, with FTIR tests confirming the decomposition of a C-H bond of $CH_2$ in linear

alkyl group in the coating material. Nevertheless, there was little remarkable difference between the behavior of the unirradiated and irradiated mortar specimens in the salt immersion test and water permeability test, indicating that UV irradiation had not completely damaged the coating. Hence, nondestructive NIRS analysis was able to identify the early signs of deterioration. We also demonstrated that the technique could identify some forms of peeling damage, based on comparison with the spectra of the individual layers of the coating material. As the analysis of each part of a structure can be completed in a short time (each measurement was completed in 1 second or less), NIRS offers the possibility of intermittent monitoring of coating deterioration. In addition, because the NIRS spectrometer is portable, it can be mounted on a drone or a robot for inspection of high-rises and other areas that are difficult to reach. These properties highlight the potential of NIRS as a simple, inexpensive method for inspection of surface coating materials.

## Supporting information

**S1 Graphical abstract.**
(TIF)

## Acknowledgments

We are grateful to Mr. Tsutomu Fukamatsu, President of Fukamatsugumi Co. LTD., for inspiring and motivating us to conduct this study. We would like to thank Professor Makoto Hisada for providing valuable advices on this work. We appreciate Mr. Hideki Fujita and Mr. Kazuo Yamamoto (Taiheiyo Consultant Co., Ltd.) for their support on measurements on mortar specimens. We also thanks to KIKUSUI Chemical Industries Co., Ltd., for telling us the general composition of the agents.

## Author Contributions

**Conceptualization:** Anri Watanabe.

**Investigation:** Anri Watanabe, Masayuki Omiya, Makoto Sato, Hiromitsu Furukawa, Nobuko Fukuda.

**Methodology:** Hiromitsu Furukawa.

**Supervision:** Hiroshi Minagawa.

**Writing – original draft:** Anri Watanabe.

**Writing – review & editing:** Hiromitsu Furukawa, Hiroshi Minagawa.

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
