## [Decision Letter · Decision Letter 0]

20 Mar 2023

PONE-D-22-27578Evaluation of near-infrared spectroscopy as a contactless method for health monitoring of coating materials applied to concrete surfacesPLOS ONE

Dear Dr. Watanabe,

Thank you for submitting your manuscript to PLOS ONE. After careful consideration, we feel that it has merit but does not fully meet PLOS ONE’s publication criteria as it currently stands. Therefore, we invite you to submit a revised version of the manuscript that addresses the points raised during the review process. Please read the comments raised by the respected reviewers and address all of them systematically.

We look forward to receiving your revised manuscript.

Kind regards,

Khalil Abdelrazek Khalil, Ph.D.

Academic Editor

PLOS ONE

Reviewers' comments:

Reviewer's Responses to Questions

**Comments to the Author**

1. Is the manuscript technically sound, and do the data support the conclusions?

Reviewer #1: Yes

Reviewer #2: No

Reviewer #3: Yes

2. Has the statistical analysis been performed appropriately and rigorously? 

Reviewer #1: Yes

Reviewer #2: N/A

Reviewer #3: N/A

3. Have the authors made all data underlying the findings in their manuscript fully available?

Reviewer #1: Yes

Reviewer #2: No

Reviewer #3: Yes

4. Is the manuscript presented in an intelligible fashion and written in standard English?

Reviewer #1: Yes

Reviewer #2: No

Reviewer #3: Yes

5. Review Comments to the Author

Reviewer #1: This article presents a good nondestructive method to track the deterioration of coatings applied to concrete surfaces. it would be of interest to many readers and i recommend accepting it after considering the following revisions:

- Lines 197-199: The authors mentioned “Given the annual amount of outdoor UV in the wavelength range of 300–400 nm, based on JIS 9110, these conditions produce deterioration equivalent to that produced by seven years of outdoor weathering”, please mention the JIS or the ASTM standards that the mentioned conditions are equivalent to “seven years of outdoor weathering”

- Line 238: please mention the JIS or the ASTM standards used with the “salt-water immersion test”.

- Lines 296-298: the authors mentioned “The reduction in the absorption intensity of the peak at 2345 nm is particularly stark compared to the peak at 2350 nm, indicating that the C-H bond is more easily broken by UV irradiation”. While in the FTIR section, lines 326-327, it is mentioned “Based on the location of these peaks, this decrease appears to indicate alkyl group cleavage”. I wonder if the results from NIR and FTIR support each others as the alkyl groups cleavage conclusion would reflect more of C-C bonds breakage rather than the C-H bonds cleavage reveled by the NIR. Please elaborate more on the results and explain more if the results from both techniques are supporting each others.

- Lines 426-427: The authors mentioned “These properties highlight the potential of NIRS as a simple, inexpensive method for inspection of surface coating materials.”. I wonder if the technique can be simply used by engineers in the filed without the need of sending some samples to labs for the SEM-EDX pictures. As it seems to me that the SEM-EDX results were also crucial to confirm the results from both the NIR and FTIR techniques, please comment on this part.

- I recommend the authors to conduct extended UV exposure beyond the 396 hours used herein as great extent of degradation might happen after 500 hours for example. Also, it would be better if the author can design a set up in which the samples get exposed to both salty water and UV simultaneously, as this combined effect might alter the coating greatly and it’s more meaningful and mimics exactly what would happen in real life, rather than the effect of UV or salt water alone.

Reviewer #2: This manuscript reports the results of an interesting research. However, I do not recommend accepting it in the present form due to various problems as described below.

The title and the short title indicate quite different contents. Judging from the title and abstract, the goal of this study is to prove that near-infrared spectroscopy can be used to evaluate the state of deterioration of coatings on concrete surfaces. However, no clear conclusion based on convincing evidence is presented. This is the main flaw of this manuscript.

Consideration on the following points is also needed.

(1) LL. 206-208: Please specify how the infrared spectrum was used to interpret the near-infrared spectrum. In this paper, near-infrared and infrared spectra are shown, respectively, in nm units and in cm-1 units, so their comparison is difficult. Use of the same units (cm-1) is recommended.

(2) LL. 281-283: Isn’t the spectrum of the underlying mortar observed not because the resin on the mortar is not even, but because the resin is so thin that near-infrared light is transmitted through it? Or was the uneven resin on the mortar concluded by the photos shown in Fig. 2?

(3) LL. 292-298: For all the spectra in Fig. 3, strong absorption of the substrate is observed suggesting that most of the coating has been lost for all samples. This is not consistent with the description in the previous paragraph.

The bands at 2345 and 2350 nm cannot be distinguished.

(4) LL. 298-301: Stating this leads that there is no basis for discussion.

(5) LL.319-320: What is the absorption at 2512 cm-1 assigned to? No significant absorption should be observed in this region usually. Is the absorption due to a component that does not change in the deterioration process and can be used for the standard?

(6) LL. 320-323: Please assign the bands clearly.

(7) LL.324-326: Please clarify which bands show similar changes by quantitative evidence.

(8) LL.327-329: Please indicate what kinds of reaction you expect.

(9) The descriptions in Sections 3.3 to 3.5 have little relation to the results of the near-infrared spectroscopy. Their relation to the main target of this manuscript cannot be understood.

(10) Only the obvious conclusion that needs no discussion is stated in section 3.6.

(11) Followings should also be revised.

･ There are many spelling mistakes, grammatical mistakes, and unnatural English expressions.

･ It is necessary to indicate what JIS is.

･ L.250: Since the SI system of units does not allow the use of M for molarity, it is recommended to use mol L-1.

･ Figs. 1, 3, 4, 9, and 10: Label for horizontal axis should be below the spectrum. It is easier to distinguish the spectra by not only distinguishing them by color, but also by adding symbols such as a, b, c, ･･･.

Reviewer #3: This paper reports the results of a study to demonstrate that NIR spectrometry can be used to inspect defects in protective coatings of concrete, including those due to ageing. The experimental design was properly set up and executed; the results are convincing and well discussed. I therefore recommend publication of this paper with some minor modifications:

- line 74: the authors mention pharmacy as a field of use of NIRS, whereas there are many others, such as synthetic chemistry (e.g. polymers), petrochemistry, food processing, etc. This passage should be amended accordingly.

- line 78: the authors write that each functional group absorbs at a unique wavelength, but this is not entirely true. True, the absorption peaks are quite sharp in the MIR, but they span several wavelengths.

- line 93: change stategic by strategic

- lines 281 - 285: the discussion suggests that the observed spectral mixing comes from lack of coverage of the protective agent. But the mixing may also be due to the fact that the light has passed through several layers of product. The few sentences in this passage could be changed in this sense, without changing the conclusion.

6. PLOS authors have the option to publish the peer review history of their article (what does this mean?). If published, this will include your full peer review and any attached files.

Reviewer #1: **Yes: **Mohammad Hassan

Reviewer #2: No

Reviewer #3: **Yes: **Jean-Michel Roger

---

## [Author Response · Author response to Decision Letter 0]

3 May 2023

We are grateful to Editor for consideration and kind support.

We wish to submit our revised manuscript and reply letter (attached) to the reviewers.

We would like to thank the reviewers for their constructive comments. We have addressed all of these in the attached document, and have revised the manuscript accordingly.

---

## [Decision Letter · Decision Letter 1]

10 May 2023

PONE-D-22-27578R1Evaluation of near-infrared spectroscopy as a contactless method for health monitoring of coating materials applied to concrete surfacesPLOS ONE

Dear Dr. Watanabe,

Thank you for submitting your manuscript to PLOS ONE. After careful consideration, we feel that it has merit but does not fully meet PLOS ONE’s publication criteria as it currently stands. Therefore, we invite you to submit a revised version of the manuscript that addresses the points raised during the review process.

We look forward to receiving your revised manuscript.

Kind regards,

Khalil Abdelrazek Khalil, Ph.D.

Academic Editor

PLOS ONE

Journal Requirements:

Reviewers' comments:

Reviewer's Responses to Questions

**Comments to the Author**

1. If the authors have adequately addressed your comments raised in a previous round of review and you feel that this manuscript is now acceptable for publication, you may indicate that here to bypass the “Comments to the Author” section, enter your conflict of interest statement in the “Confidential to Editor” section, and submit your "Accept" recommendation.

Reviewer #3: All comments have been addressed

Reviewer #4: (No Response)

2. Is the manuscript technically sound, and do the data support the conclusions?

Reviewer #3: (No Response)

Reviewer #4: Partly

3. Has the statistical analysis been performed appropriately and rigorously? 

Reviewer #3: (No Response)

Reviewer #4: I Don't Know

4. Have the authors made all data underlying the findings in their manuscript fully available?

Reviewer #3: (No Response)

Reviewer #4: Yes

5. Is the manuscript presented in an intelligible fashion and written in standard English?

Reviewer #3: (No Response)

Reviewer #4: Yes

6. Review Comments to the Author

Reviewer #3: (No Response)

Reviewer #4: Manuscript Number: PONE-D-22-27578R1

Evaluation of near-infrared spectroscopy as a contactless method for health monitoring of coating materials applied to concrete surfaces

The work proposes using NIR spectroscopy for characterization of coating materials applied to concrete. I suggest the authors to present what are these materials (resin-based materials) in the abstract. Also, perhaps this can be added to the title, to clarify to the audience the materials investigated. This would be important, considering the general knowledge (and also what the authors mention in the introduction) about the signals in the NIR spectra that are related to organic material.

Material and methods

Section 2.1. Please clarify the total number of samples investigated. Also, please clarify the number of samples used for calibration and validation, as it is mandatory to have an external validation dataset.

Results and discussion

As the authors present, the main test for the degradation was the water-permeability, and water has a signal on NIR spectra. Hence, it seems that the main determination is actually the presence of water in the coatings. I suggest the authors to clarify this information, and enhance the discussion, considering this aspect.

Previous works have used NIRS to determine the mechanical properties of different materials. However, the mechanical properties are correlated to the composition of organic materials. Hence, I hereby list a few works on the subject (as a suggestion, the authors may find others) that may provide support for the discussion, stating that the main determination is actually the composition or presence of water, and it is correlated to the coating materials deterioration.

VIS–NIR spectroscopy as a process analytical technology for compositional characterization of film biopolymers and correlation with their mechanical properties - https://doi.org/10.1016/j.msec.2015.06.029

Application of VIS/NIR spectroscopy for estimating chemical, physical and mechanical properties of cork stoppers. Wood Sci Technol 48, 811–830 (2014). https://doi.org/10.1007/s00226-014-0642-3

Comparison of Methods for Estimating Mechanical Properties of Wood by NIR Spectroscopy - https://doi.org/10.1155/2018/4823285

I suggest to change the short title presented in line 4, page 1/58

7. PLOS authors have the option to publish the peer review history of their article (what does this mean?). If published, this will include your full peer review and any attached files.

Reviewer #3: **Yes: **Jean-Michel Roger

Reviewer #4: No

---

## [Author Response · Author response to Decision Letter 1]

13 Jun 2023

Reply to Reviewers

Reviewer #4: Manuscript Number: PONE-D-22-27578R1

Evaluation of near-infrared spectroscopy as a contactless method for health monitoring of coating materials applied to concrete surfaces

The work proposes using NIR spectroscopy for characterization of coating materials applied to concrete. I suggest the authors to present what are these materials (resin-based materials) in the abstract. Also, perhaps this can be added to the title, to clarify to the audience the materials investigated. This would be important, considering the general knowledge (and also what the authors mention in the introduction) about the signals in the NIR spectra that are related to organic material.

-Thank you for your kind suggestions. We inserted the words “resin-based” and “organic resin-based” for title and abstract.

Previous Title: Evaluation of near-infrared spectroscopy as a contactless method for health monitoring of coating materials applied to concrete surfaces 

Revised Title: Evaluation of near-infrared spectroscopy as a contactless method for health monitoring of resin-based coating materials applied to concrete surfaces 

Previous Line 27: simple inspection for health monitoring of coating materials.

Revised Line 27: simple inspection for health monitoring of organic resin-based coating materials.

Material and methods

Section 2.1. Please clarify the total number of samples investigated. Also, please clarify the number of samples used for calibration and validation, as it is mandatory to have an external validation dataset.

-We prepared a single sample for each measurement, since the space of the sample holder of the UV irradiation equipment is limited. We revised the manuscript as follows. We also added the information for Section 3.6, without UV irradiation.

Added line 156: Totally, 2 (before/after UV irradiation) × 4 (coating thickness) × 5 (kinds of measurements, in case of NIRS, FTIR, Permeability test, Salt-water immersion test, SEM) = 40 samples were prepared.

Added line 180: The total number of the samples for the peeling characterization was four. 

In this paper, we did not use statistical method such as chemometrics yet. Since this study is the fundamental investigation to clarify the ability of deterioration detection by NIRS, when the degree of the deterioration is slight.

Results and discussion

As the authors present, the main test for the degradation was the water-permeability, and water has a signal on NIR spectra. Hence, it seems that the main determination is actually the presence of water in the coatings. I suggest the authors to clarify this information, and enhance the discussion, considering this aspect.

-It is well known that the protective organic paintings are deteriorated by mainly UV irradiation during weathering test. This photo-oxidative deterioration decreases water protection ability; therefore the permeability of organic coatings generally increases with longer period of exposure to UV irradiation [“Influence of the photo-oxidative degradation on the water barrier and corrosion protection properties of polyester paints”, F. Deflorian, L. Fedrizzi, P.L. Bonora, Corrosion Science, vol. 38 (1996) 1697-1708]. [“Influence of coating system composition on moisture dynamic performance of coated wood,” Jan Ekstedt, Journal of Coatings Technology vol. 75 (2003) 27–37]. 

In this study, it was found that the chemical deterioration due to moderate UV irradiation did not affect permeability yet. Since we haven’t investigated the physical degradation such as cracks and peelings caused by this chemical deterioration with more harsh UV irradiation condition, it will be confirmed in the next study.

As the Reviewer suggests, the absorption peak originated from “-OH of water” clearly can be seen in Fig 1 and Fig 3. The reason why these the peak intensity different between before and after UV irradiation, may be because large amount of water was applied to the samples by the shower during the weathering test or because some chemical reaction changed the environment around -OH bond. We think that the former possibility is reasonable, since all of the peaks at around 7000 cm-1 and 5200 cm-1 radically and similarly enhanced in thin, normal, and thick coating samples in Fig. 3.

As our future task, we would like to investigate the easier method to evaluate the deterioration level, which is directly related to protection ability of the coating materials, by carefully referring results of more through comparison experiments.

 We added the following sentences to the manuscript,

Line 234: It is well known that the protective organic paintings are deteriorated by mainly UV irradiation during weathering test. This photo-oxidative deterioration decreases water protection ability; therefore, the permeability of organic coatings generally increases with longer period of exposure to UV irradiation [32][33].

Line 355: It is noteworthy that the prominent braod peaks of water appeared in Fig. 3 at around 5200 cm-1 and 7000 cm-1, which may be because the large amount of water was applied to the samples by the shower during the weathering test. 

Previous works have used NIRS to determine the mechanical properties of different materials. However, the mechanical properties are correlated to the composition of organic materials. Hence, I hereby list a few works on the subject (as a suggestion, the authors may find others) that may provide support for the discussion, stating that the main determination is actually the composition or presence of water, and it is correlated to the coating materials deterioration.

VIS–NIR spectroscopy as a process analytical technology for compositional characterization of film biopolymers and correlation with their mechanical properties - https://doi.org/10.1016/j.msec.2015.06.029

Application of VIS/NIR spectroscopy for estimating chemical, physical and mechanical properties of cork stoppers. Wood Sci Technol 48, 811–830 (2014). https://doi.org/10.1007/s00226-014-0642-3

Comparison of Methods for Estimating Mechanical Properties of Wood by NIR Spectroscopy - https://doi.org/10.1155/2018/4823285

 -This study is the previous stage of the estimating physical/chemical properties quantitatively. Therefore, only qualitative discussions are written. However, thank you very much for suggesting us to try the quantitative prediction of the deterioration degree. We will prepare many samples enough for the statistical calibration such as PLS in the next study. At the first onset, we have just prepared another new UV irradiation equipment and made new sample holders which we can put more samples at once. We added the sentence as follows:

Line 78: Also, visible and NIR spectroscopy is often used for prediction of physical and mechanical properties, supported by chemometrics procedures [23][24][25].

I suggest to change the short title presented in line 4, page 1/58

-We revised the short title, following to the advice of the Reviewer.

Previous short title: UV does not degrade the mass transfer resistance of concrete coatings.

Revised short title: Moderate UV irradiation does not degrade the mass transfer resistance of concrete covered with organic resin.

---

## [Decision Letter · Decision Letter 2]

15 Jun 2023

Evaluation of near-infrared spectroscopy as a contactless method for health monitoring of resin-based coating materials applied to concrete surfaces

PONE-D-22-27578R2

Dear Dr. Watanabe,

We’re pleased to inform you that your manuscript has been judged scientifically suitable for publication and will be formally accepted for publication once it meets all outstanding technical requirements.

Kind regards,

Khalil Abdelrazek Khalil, Ph.D.

Academic Editor

PLOS ONE

Additional Editor Comments (optional):

Reviewers' comments:

Reviewer's Responses to Questions

**Comments to the Author**

1. If the authors have adequately addressed your comments raised in a previous round of review and you feel that this manuscript is now acceptable for publication, you may indicate that here to bypass the “Comments to the Author” section, enter your conflict of interest statement in the “Confidential to Editor” section, and submit your "Accept" recommendation.

Reviewer #4: All comments have been addressed

2. Is the manuscript technically sound, and do the data support the conclusions?

Reviewer #4: (No Response)

3. Has the statistical analysis been performed appropriately and rigorously? 

Reviewer #4: (No Response)

4. Have the authors made all data underlying the findings in their manuscript fully available?

Reviewer #4: (No Response)

5. Is the manuscript presented in an intelligible fashion and written in standard English?

Reviewer #4: (No Response)

6. Review Comments to the Author

Reviewer #4: (No Response)

7. PLOS authors have the option to publish the peer review history of their article (what does this mean?). If published, this will include your full peer review and any attached files.

Reviewer #4: No

---

## [Editor Report · Acceptance letter]

20 Jun 2023

PONE-D-22-27578R2 

Evaluation of near-infrared spectroscopy as a contactless method for health monitoring of resin-based coating materials applied to concrete surfaces 

Dear Dr. Watanabe:

I'm pleased to inform you that your manuscript has been deemed suitable for publication in PLOS ONE. Congratulations! Your manuscript is now with our production department. 

Kind regards, 

on behalf of

Dr. Khalil Abdelrazek Khalil 

Academic Editor

PLOS ONE